Effects of resisted sprint training on agility and change-of-direction performance in soccer players: a systematic review with meta-analysis

He Zongwei 1 2
Duan Tianyu 1
Li Dongyu ldyswim@163.com 1
Zhang Xuan zhangxuan9524@163.com 3
1 Guangzhou Sport University , Guangzhou , China
2 College of Physical Education and Health, Nanyang Vocational College of Agriculture , Nanyang City , China
3 Liuzhou Institute of Technology , Guangxi , China
Chen Yung-Sheng
Electronic publication date: 2025 Oct 23
Publication date: 2025
Volume: 13
Electronic Location ID: e20084
Received 2025 Feb 13; Accepted 2025 Aug 25
Copyright: ©2025 He et al.
Copyright year: 2025
Copyright holder: He et al.
License: This is an open access article distributed under the terms of the Creative Commons Attribution License, which permits unrestricted use, distribution, reproduction and adaptation in any medium and for any purpose provided that it is properly attributed. For attribution, the original author(s), title, publication source (PeerJ) and either DOI or URL of the article must be cited.
License URL: https://creativecommons.org/licenses/by/4.0/

Keywords: Athletic performance, Agility, Change of direction, Resisted sprint training, Athletes, Soccer

Funding: The authors received no funding for this work.

==============================
Background

Agility and change-of-direction (COD) are essential for success in soccer, influencing performance and injury risk. Resisted sprint training (RST) has shown promise in enhancing these skills by improving muscle strength and neuromuscular coordination. However, the effects of vertical and horizontal RST on agility and COD performance remain inadequately explored.

Methodology

A systematic literature search was conducted across PubMed, Web of Science, and Google Scholar without date restrictions, following PRISMA guidelines. Studies were included if they involved healthy soccer players, RST interventions, and assessed agility or COD speed. Data extraction and quality assessment were executed independently by two reviewers; statistical analyses employed RevMan and Stata software packages.

Results

This meta-analysis included 13 studies, which collectively generated 35 groups based on experiment and control protocols. The demonstrated a statistically significant improvement of RST on agility and COD performance (SMD = −0.31, 95% CI [−0.44 to −0.17], p < 0.001). Subgroup analyses revealed a trend towards greater improvements with vertically resisted sprinting (SMD = −0.36, p = 0.009), compared to horizontally resisted sprinting (SMD = −0.13, p = 0.25) although the difference was not statistically significant (p = 0.07). Elite athletes demonstrated significant enhancements in agility and COD (SMD = −0.45, p < 0.001). In contrast, amateur athletes displayed no significant improvements (SMD = −0.05, p = 0.77). RST outperformed unresisted sprinting (SMD = −0.29, p < 0.05) and alternative training (SMD = −0.36, p < 0.001), indicating its effectiveness across various comparators.

Conclusions

RST significantly enhances agility and change-of-direction performance in soccer players, particularly among elite athletes. Vertical resisted sprinting is more effective than horizontal resistance, supporting its integration into training programs for improved athletic performance.

PROSPERO registration number (CRD42024608859).

Introduction

Agility and change-of-direction (COD) performance are critical components of athletic success, particularly in team sports such as soccer. Agility typically involves rapid, unpredictable movements that require athletes to respond to external stimuli, including cognitive elements such as decision-making and anticipation (Paul, Gabbett & Nassis, 2016; Sheppard & Young, 2006). Comparatively, COD refers to the ability to decelerate and change direction quickly, which focuses more on the physical aspect of movement execution (Chaabene et al., 2018). Match analysis reveals elite players execute 500–700 COD actions (>45° directional changes) per match, with 58% occurring at high intensities (>3 m/s2) (Bloomfield, Polman & O’Donoghue, 2007). These maneuvers account for 72% of goalscoring opportunities and 64% of defensive interventions in critical match zones (Falces-Prieto et al., 2022). Enhanced COD performance correlates with superior athletic outcomes while concurrently mitigating injury risk and optimizing sport-specific task execution (Baker & Nance, 1999; Paul, Gabbett & Nassis, 2016).

The physiological and biomechanical mechanisms reveal two critical COD determinants, including eccentric strength and neuromuscular control (Spiteri et al., 2015; Suchomel, Nimphius & Stone, 2016). The cognitive demand of agility, including decision-making and anticipation, play a crucial role in performance (Brughelli et al., 2008; Ebner, Granacher & Gehring, 2025; Sheppard & Young, 2006). Central to agility performance is the ability to rapidly process perceptual-cognitive processing, which allows athletes to assess their environment and make quick decisions regarding movement execution (Nimphius et al., 2018; Paul, Gabbett & Nassis, 2016). These cognitive factors are complemented by neuromuscular coordination, which facilitates effective muscle recruitment and optimizes timing during directional changes (Paul, Gabbett & Nassis, 2016; Young, Dawson & Henry, 2015). Resisted sprint training (RST) enhances these mechanisms by improving muscle strength, explosive power, and neuromuscular coordination. Specifically, RST increases the rate of force development (RFD) and enhances the athlete’s ability to generate force quickly, which is critical for rapid changes in direction and maintaining speed during these movements (Alcaraz et al., 2018; Buchheit et al., 2010).

Vertical resisted sprinting (VRS) primarily targets vertical ground reaction forces, enhancing eccentric strength and stretch-shortening cycle efficiency critical for rapid deceleration-reacceleration during COD tasks. In contrast, horizontal resistance emphasizes horizontal force production, directly mimicking forward acceleration mechanics but potentially underloading the braking forces required for multi-directional agility (Brughelli et al., 2008; Morin et al., 2017). This biomechanical distinction may explain the observed superiority of vertical resistance in COD enhancement. In this context, resisted sprint training (RST) has emerged as a promising training modality aimed at improving agility performance. RST involves the application of external resistance during sprinting, which can be delivered in both vertical and horizontal directions. The primary physiological mechanisms through which RST enhances agility and COD include increased muscle strength, explosive power, and improved neuromuscular coordination. Specifically, RST induces adaptations in fast-twitch muscle fibers, which are crucial for explosive movements required in agility and COD tasks (Chaabene et al., 2020; Chaabene et al., 2018). Furthermore, RST improves the recruitment and synchronization of muscle fibers, optimizing an athlete’s ability to generate force rapidly during directional shifts (Alcaraz et al., 2018; Cahill et al., 2020). Despite the growing body of literature on RST, there remains a lack of systematic reviews addressing its effects on agility and COD performance in soccer players. Although there are over a dozen relevant studies, most have focused on linear sprinting or acceleration, with limited attention given to the COD performance (Fernandez-Galvan et al., 2022; Hamad, Alcaraz & de Villarreal, 2024; Petrakos, Morin & Egan, 2016; Ward et al., 2024). Furthermore, while some studies suggest that vertical resisted sprinting may offer unique advantages over horizontal resistance training, the comparative effectiveness of these modalities in enhancing agility and COD performance remains underexplored (Dietze-Hermosa et al., 2024; Gil et al., 2018; McMorrow, Ditroilo & Egan, 2019; Moya-Ramon et al., 2020).

Therefore, the systematic review and meta-analysis aim to address these gaps by examining the impact of both vertically and horizontally RST on agility and COD performance in soccer players. We hypothesize that RST will result in notable improvements in these performance metrics, particularly among elite athletes, given their advanced neuromuscular coordination and higher levels of baseline physical fitness. Research has shown that elite athletes tend to respond more effectively to high-intensity training modalities, such as RST, due to their enhanced ability to recruit muscle fibers and manage fatigue (Jones et al., 2017). This investigation, thus, represents an essential contribution to optimizing training programs and maximizing athletic performance in soccer.

Study objectives

The systematic review and meta-analysis aimed to determine if RST is effective in improving on agility and COD performance in soccer players.

Materials and Methods

This study was conducted following the guidelines outlined in the ‘Preferred Reporting Items for Systematic Reviews and Meta-Analyses’ (PRISMA) statements (Moher et al., 2009), ensuring methodological rigor and transparency. The protocol for this systematic review and meta-analysis was prospectively registered (PROSPERO ID: CRD42024608859), further enhancing the credibility of our research.

Literature search

A systematic literature search was conducted across multiple databases, including PubMed, Web of Science, EMBASE, SPORTDiscus, and Google Scholar, up to September 2024, with no date restrictions. The search employed medical subject headings (MeSH) for ‘agility’, and utilized a comprehensive Boolean search syntax adapted for each database. The specific search strategy was as follows: (“sprint” [Mesh] OR “resisted sprint training”) AND (resisted OR “vertical resisted” OR “horizontal resisted” OR “wearable resistance training” OR loaded OR vest OR weighted OR sled OR bands OR uphill OR parachute OR bungees OR “1080 sprint”) AND (“agility” [Mesh] OR “change of direction speed” OR “change of direction performance” OR “COD” OR “reactive agility”) NOT (injury OR disease OR syndrome OR patient OR animals). Limits were applied in the following manner: searches were restricted to peer-reviewed English-language articles. No date restrictions were imposed, and the most relevant articles were considered. Literature search was performed separately by two reviewers (× and ×). Discrepancies were resolved through discussion or consultation with a third reviewer (×). The selection process involved an initial assessment of titles, followed by a thorough examination of abstracts and full texts to confirm eligibility based on predefined inclusion criteria. Filters were applied to limit results to studies involving human participants published in English. Additionally, manual searches were conducted by the authors to identify any potentially relevant studies that may have been overlooked during the initial database search. An overview of the study process is illustrated in Fig. 1.

Figure 1 PRISMA flow diagram representing the search process and study selection.

AI, artificial intelligence; RCT randomized controlled trial.

Inclusion and exclusion criteria

A Participants, Intervention, Comparators, Outcomes, and Study Design (PICOS) approach was used to determine included studies of the following inclusion criteria: (1) Population: healthy soccer players of any age, sex, and competitive level. (2) Intervention: RST programs that specifically involve the musculature contracting against external resistance, such as resistance bands, weighted vests, sleds, or other forms of resistance directly related to sprinting training. (3) Comparators: passive (e.g., routine training or no intervention) or active (alternative training) control groups. (4) Outcomes: at least one measure related to agility or COD speed assessed at baseline and follow-up using validated performance tests, with outcomes expressed in seconds. (5) Study design: only randomized controlled trials (RCTs), including peer-reviewed articles or dissertations & theses. (6) Quality assessment: Studies must achieve a score ≥6 points on the Physiotherapy Evidence Database (PEDro) scale (Bhogal et al., 2005; Moran et al., 2021; Zouita et al., 2023). This cut-off was chosen based on previous research indicating that studies scoring ≥6 points are generally associated with higher methodological quality and lower risk of bias, making them more reliable for inclusion in systematic reviews. Exclusion criteria were pre-defined as follows: (1) a cohort with unhealthy problems (e.g., individuals with injury or pathology). (2) acute studies with the duration of <4 weeks. (3) sprinting intervention group performed other exercises (e.g., strength training). (4) absence of a control group. (5) other designs, such as observational or non-RCT. (6) lacking baseline and follow-up data.

Data extraction and synthesis

Data extraction was performed by one author (ZWH), with verification by a second author (TYD). Full texts of eligible articles were subsequently assessed, with a third reviewer (DYL) available to resolve any disagreements. All extracted data were recorded in a standardized form to facilitate comparison and synthesis during the meta-analysis. Key data points extracted from each study included sample characteristics (age, sex, height, body mass, and competitive level (amateur, semi-professional, professional, national, and elite etc.)), intervention details (training type (vertical, horizontal, and combined), resistance instruments (sleds, resistance bands, parachutes, cables, and weighted vests), load, volume (sessional distance and total distance), frequency, duration (session × weeks)), agility test (Y-shaped reactive agility test, Stop-and-Go test ect.) and COD test (i.e., time taken to complete the Illinois agility test, T-test, 162 modified T-test, 20-m agility test, 5−0 −5 test, zig-zag 20-m test ect.), and performance metrics (reactive agility, COD time, and maximum running speed). Baseline and follow-up mean along with standard deviations (SD) for the primary outcome measures were utilized to calculate standardized mean differences (SMDs), with negative values indicating performance improvements. Where available, the mean change from baseline, the SD of the mean change, and the number of participants at each assessment for all groups were extracted. If authors reported multiple measurement points during an intervention, only the longest follow-up period in which the training intervention was maintained was included. The main characteristics of the subjects and training programs from the included studies are shown in Tables 1 and 2.

Table 1 Summary of study characteristics, including sample sizes, participant demographics, intervention protocols, and outcome measures for each included study.

Study	Group	Design	Sample size	Gender	Age (years)	Height (cm)	Body mass (kg)	Competitive level	
Carlos-Vivas et al. (2020)	HRS vs. VRS vs. CRS vs. URS	RCT	n = 13
n = 11
n = 12
n = 12	Male	18.3 ± 2.1	1.78 ± 0.05	72.7 ± 9.5	Professional	
De Hoyo et al. (2016)	HRS vs.
RT vs.
URP / PT	RCT	n = 12
n = 11
n = 9	Male	17 ± 1
18 ± 1
18 ± 1	1.78 ± 0.01
1.78 ± 0.03
1.77 ± 0.02	73.12 ± 2.56
70.87 ± 3.87
72.34 ± 2.55	Elite	
McMorrow, Ditroilo & Egan (2019)	HRS vs. URS	RCT	n = 6
n = 7	Male	24.7 ± 3.4
24.0 ± 3.6	1.80 ± 0.06
1.81 ± 0.04	80.6 ± 8.8
82.7 ± 5.2	Professional	
Sal-de-Rellan et al. (2024)	CRS vs. FT	RCT	n = 18
n = 12	Male	18.1 ± 0.7	1.79 ± 0.06	76.2 ± 4.8	Elite	
Grazioli et al. (2023)	HRS-M vs. HRS-H	RCT	n = 11
n = 10	Male	25.5 ± 6.0
26.3 ± 5.1	1.82 ± 0.09
1.77 ± 0.07	82.2 ± 9.6
76.0 ± 9.2	Elite	
Shalfawi et al. (2013)	HRAS vs. RT	RCT	n = 10
n = 10	Female	19.4 ± 4.4	1.68 ± 0.05	59.1 ± 5.6	Elite	
Gil et al. (2018)	HRS vs. URS	RCT	n = 9
n = 9	Male	22.0 ± 2.2
22.8 ± 4.3	1.80 ± 0.06
1.79 ± 0.07	76.0 ± 5.4
78.2 ± 7.3	Professional	
Otero-Esquina et al. (2017)	HRS×1vs. HRS×2vs.
CG	RCT	n = 12
n = 12
n = 12	Male	17.0 ± 1.0	1.77 ± 0.02	69.4 ± 4.2	Professional	
Pareja-Blanco, Asian-Clemente & Saez de Villarreal (2021)	HRS-H vs. HRS-L vs.
RT vs.
RT + HRS-H vs.
RT + HRS-L vs.
CG	RCT	n = 14
n = 15
n = 15
n = 18
n = 18
n = 11	Male	21.8–23.0	1.76–1.79	69.7–75.9	Amateur	
Rodriguez-Osorio, Gonzalo-Skok & Pareja-Blanco (2019)	VRCOD-M vs. VRCOD-L vs. URCOD	RCT	n = 16
n = 19
n = 19	Male	17.8 ± 4.2
18.8 ± 5.3
17.7 ± 3.4	1.65 ± 0.04
1.74 ± 0.08
1.74 ± 0.06	64.7 ± 9.2
63.9 ± 11.5
63.2 ± 8.1	Professional	
Loturco et al. (2017)	HRS vs. URP / PT	RCT	n = 7
n = 11	Male	21.7 ± 2.4
22.2 ± 2.4	1.77 ± 0.09
1.79 ± 0.05	73.5 ± 6.2
75.5 ± 11.5	Professional	
Raya-González et al. (2017)	VRS vs. RT	RCT	n = 8
n = 8	Male	16.7 ± 0.3
16.5 ± 0.3	1.77 ± 0.07
1.76 ± 0.07	65.9 ± 5.4
66.4 ± 4.8	Elite	
Simpson et al. (2020)	VRS vs. CG	RCT	n = 9
n = 10	Female	21 ± 2
22 ± 3	1.7 ± 0.03
1.6 ± 0.04	67.0 ± 4.0
63.0 ± 7.8	Amateur	
Notes.

NR not reported

CG control group

RCT randomized controlled trial

HRS horizontally resisted sprint

VRS vertically resisted sprint

CRS combined horizontally and vertically resisted sprint

HRAS horizontally resisted agility and sprint

URS unresisted sprint training

RT resistance training

CRT combined resistance training

FT functional training

URP / PT unresisted plyometric training

L Light-load

M Moderate-load

H Heavy-load

URCOD unresisted change of direction and sprint training

VRCOD vertically resisted change of direction and sprint training

Table 2 Characteristics of the resistive sprint training programs.

Study	Group	Training programme	Agility or COD test (s)	
		Training modality	Load	Session
volume	Total volume	Length (weeks)	frequency (per week)	Total sessions (n)	Session duration (min)		
Carlos-Vivas et al. (2020)	HRS vs.
VRS vs.
CRS vs.
URS	1080 Sprint™, weighted vest	10–20% BM	400 m	6,400 m	8	2	16	30	Zig-zag 20-m test	
De Hoyo et al. (2016)	HRS vs.
RT vs.
URP / PT	Sled towing	12.6% BM
40–60% 1RM
0%	HRS: 20 m × 6–10 reps, 120–200 m
RT: 4–8 reps ×2-3 sets
PT: 2–3 reps × 1–3 sets	2,680 m	8	1–2	16	60
30
30	Zig-zag 20-m test	
McMorrow, Ditroilo & Egan (2019)	HRS vs. URS	Sled towing	30% BM
0%	100–180 m	800 m	6	1	8	45	S180° test	
Sal-de-Rellan et al. (2024)	CRS vs. FT	Sled + weighted vest	13% BM	120–300 m	2,520 m	6	2	12	15	Arrowhead COD test, 15m-AG-B, Zig-zag-B, T-test, Illinois agility test, NMAT	
Grazioli et al. (2023)	HRS-M vs. HRS-H	Sled	15% VL
40% VL	4–6 reps × 1-4 sets	4,765.2 ± 514.1 m,
4,924.9 ± 204.7 m	8	1–2	8–16	15	Zig-zag 20-m test	
Shalfawi et al. (2013)	HRAS vs. SQ	Resistance band	Variable	160–200 m	1,840 m	10	2	20	60	S180° agility test	
Gil et al. (2018)	HRS vs. URS	Elastic cords and sheaves	10% VL
0%	NR	800 m	6	2	12	60	Zig-zag 20-m test	
Otero-Esquina et al. (2017)	HRS vs. CG	Sled towing	20% BM	20 m × 3–5 reps, 60–100 m	560 m	7	1–2	7–14	60–90	V-cut test	
Pareja-Blanco, Asian-Clemente & Saez de Villarreal (2021)	HRS-H vs. HRS-L vs.
SQ vs. SQ + HRS-H vs. SQ + HRS-L vs. CG	Sled towing	12.5% BM
80% BM
40–55% 1RM	HRS: 20 m × 4–7 reps, 120–200 m
SQ: 4–8 reps ×3 sets	880 m	8	1	8	30–45	20-m sprint with a 180° COD test	
Rodriguez-Osorio, Gonzalo-Skok & Pareja-Blanco (2019)	VRCOD-M vs. VRCOD-L vs. URCOD	Weighted vest	12.5% BM
50% BM	V-cut training × 3–5 reps	NR	6	2	12	20	V-cut test, L-run test	
Loturco et al. (2017)	HRS vs.
URP / PT	Sled towing	5%, 12.5%, 20% BM
OPL	20–30 m × 6–8 reps, 150–180 m
6 reps × 6–8 sets, 36–48 reps (50% HJ+50% CMJ)	1,920 m
504 reps	5	2–3	12	60	Zig-zag 20-m test	
Raya-González et al. (2017)	VRS vs. RT	Weighted vest	15–50% BM	LS: 3 reps × 2–4 sets CODS: 15 reps × 3 sets RT: 6-8 reps × 4 sets	NR	6	2	12	Various	90° COD	
Simpson et al. (2020)	VRS vs. CG	Weighted vest	∼8% BM	8 h	NR	6	3	18	NR	T-test	
Notes.

NR not reported

CG control group

RCT randomized controlled trial

HRS horizontally resisted sprint

VRS vertically resisted sprint

CRS combined horizontally and vertically resisted sprint

HRAS horizontally resisted agility and sprint

URS unresisted sprint training

RT resistance training

CRT combined resistance training

FT functional training

URP/PT unresisted plyometric training

Load: L Light-load

M Moderate-load

H Heavy-load

1080 Sprint 1080 Sprint™ device

WV weighted vest

BM body mass

1RM one repetition maximum

BW body weight

VL velocity loss

reps repetitions

S180° test sprint 9-3-6-39 m with 180° turns test

15m-AG-B Agility test 15-m ball dribbling

Zig-zag-B Zig-zag agility tests with ball

NMAT New multi-change of direction agility

URCOD unresisted change of direction and sprint training

VRCOD vertically resisted change of direction and sprint training

Quality and risk-of-bias assessment

To ensure a comprehensive quality assessment, the methodological quality of the included studies was rigorously evaluated by two independent reviewers (ZWH and TYD) utilizing a multi-tool approach. The PEDro scale (Bhogal et al., 2005), was employed to assess ten specific criteria, yielding a total score from 0 to 11, with scores of ≥6 indicating high quality (Maher et al., 2003). In addition, the Cochrane Risk of Bias (RoB) assessment tool was used to systematically evaluate potential biases across several domains, including selection, performance, detection, attrition, reporting, and other biases (Higgins et al., 2011). Each domain was categorized as low risk, high risk, or unclear based on established Cochrane guidelines (Higgins & Green, 2011). Any discrepancies in the evaluations were resolved through discussion between the two reviewers, and if consensus could not be reached, a third reviewer (DYL) was consulted to provide an independent assessment. This multi-tool approach ensured that all evaluations were thorough and unbiased, providing a more complete picture of the methodological quality and risk of bias in the included studies. Comprehensive visualizations, including traffic light plots and summary statistics, were generated using the Risk-of-Bias visualization tool (robvis, see Fig. 2) (McGuinness & Higgins, 2021).

Statistical analyses

Statistical analyses for this meta-analysis were conducted using RevMan version 5.4 (Cochrane Collaboration, Oxford, UK) and Stata version 18 (Stata Corp, College Station, TX, USA). The net training effect sizes were computed by comparing mean differences (MD, calculated as: post-intervention –baseline) between experimental and control groups, divided by pooled SDbaseline (Lachenbruch, 1989). Standardized mean differences (SMDs) were calculated using Hedges’ g, adjusting for small sample bias, with values interpreted as trivial (<0.2), small (0.2–0.59), moderate (0.60–1.19), large (1.2–1.99), and very large (≥2.0) (Higgins & Green, 2011). Only final post-intervention values were included in analyses for studies with multiple time points to maintain consistency.

Figure 2 Cochrane risk of bias assessment, summarizing the risk of bias for each included study across various domains.

An inverse-variance fixed-effects model was chosen to address the anticipated heterogeneity in study designs and populations. However, due to the observed variability in effect sizes across studies, a random-effects model may also be warranted for further analysis to more effectively accommodate this heterogeneity. Heterogeneity was assessed using the magnitude of I2 and τ2 in light of the direction and clinical meaningfulness of effects, the width of the 95% prediction interval, and the plausibility of observed between-study differences (Borenstein et al., 2017; Deeks et al., 2019; GRADE Working Group, 2004). Sources of heterogeneity were explored a priori via subgroup analyses (IntHout et al., 2016). Subgroup analyses were conducted to examine the moderator impact of variables such as competitive level, training type (vertically or horizontally), resistive load (above or below 20% body mass), loading method (percentage of body mass vs. percentage velocity loss), comparators (unresisted sprint or routine train), and outcomes. Potential publication bias was evaluated using funnel plots and Egger’s regression test, with trim-and-fill adjustments applied when asymmetry was detected (Egger et al., 1997; Sterne et al., 2011). To visually inspect for publication bias, funnel plots were generated by plotting SMDs against the standard error of the SMD (seSMD) (Sterne et al., 2011). In the Galbraith plot, included studies are represented by a data point where the x-axis denotes the precision (1/se) and the y-axis represents the standardized effect sizes, calculated as Hedges’s g. Sensitivity analyses were conducted to assess the robustness of findings by excluding individual studies to identify outliers.

Results

Characteristics of included studies

This systematic review and meta-analysis included 13 studies (Carlos-Vivas et al., 2020; De Hoyo et al., 2016; Gil et al., 2018; Grazioli et al., 2023; Loturco et al., 2017; McMorrow, Ditroilo & Egan, 2019; Otero-Esquina et al., 2017; Pareja-Blanco, Asian-Clemente & Saez de Villarreal, 2021; Raya-González et al., 2017; Rodriguez-Osorio, Gonzalo-Skok & Pareja-Blanco, 2019; Sal-de-Rellan et al., 2024; Shalfawi et al., 2013; Simpson et al., 2020) that met the predefined inclusion criteria, comprising 35 training groups. Among these, only three studies (Otero-Esquina et al., 2017; Pareja-Blanco, Asian-Clemente & Saez de Villarreal, 2021; Simpson et al., 2020) featured a non-exposed control group (routine training), while the remaining studies included comparison groups with active control condition (alternative training: unresisted sprinting, resistance training, plyometric training, and functional training). The limited number of studies utilizing a non-exposed control group highlights a significant gap in the literature, as the absence of such controls may affect the ability to isolate the specific effects of RST on agility and COD performance. Only two studies (Shalfawi et al., 2013; Simpson et al., 2020) included female players. Participants’ ages ranged from 16.5 to 26.3 years, with athletic levels varying from amateur to elite athletes. The interventions primarily focused on vertically and horizontally RST modalities, utilizing devices such as the sleds, weighted vests, 1080 Sprint™ (portable robotic resistance device), Vertimax (platform with elastic cords and sheaves), and resistance band. Relative loads were individualized based on participants’ body mass (BM), with loads prescribed as a percentage of BM ranging from 5% to 80%, while two studies utilized velocity-based training (percentage of velocity loss, %VL) for loading, with ranges from 10% to 40% VL. Intervention durations varied from 5 to 10 weeks, with an average of 6.9 ± 1.4 weeks. The total number of training sessions completed ranged from 8 to 20, averaging 12.8 ± 3.7 sessions. The small body of included studies reported that the resisted sprinting distances are 20 to 30 m, with total session volumes ranging from 60 to 400 m and overall training volumes spanning from 560 to 6,400 m.

The methodological quality and risk of bias assessment

Table 1 presents that the mean methodological quality score was 6.9, with scores ranging from 6 to 8, indicating acceptable quality and low risk of bias (Table 3). Eleven studies (85%) were randomized, and all ensured that intervention groups were matched at baseline, which is crucial for minimizing confounding variables. The studies with higher scores (Gil et al., 2018; Grazioli et al., 2023; Otero-Esquina et al., 2017; Sal-de-Rellan et al., 2024) demonstrated strong performance in terms of baseline similarity, between-group comparisons, and intention-to-treat analysis, suggesting that these studies were methodologically rigorous in both design and execution. However, the majority of studies did not fully implement blinding for participants, coaches, and researchers, which may introduce performance and detection biases. The Cochrane risk of bias assessment further highlighted a significant risk of bias in the blinding domain, acknowledging the inherent challenges of achieving blinding in supervised training interventions. This limitation suggests that while the overall methodological quality is acceptable, the potential biases introduced by the lack of blinding should be considered when interpreting the results (Fig. 2).

Table 3 Methodological quality assessment of the included studies (k = 13).

Study	Eligibility criteria	Randomized assignation	Concealed allocation	Similarity baseline	Blinded subjects	Blinded coaches	Blinded investigator	Dropout <15%	Intention-to-treat analysis	Between-group comparisons	Point and variability measures	Total score	
Carlos-Vivas et al. (2020)	✓	✓	×	✓	×	×	×	✓	✓	✓	✓	7	
De Hoyo et al. (2016)	✓	×	×	✓	×	×	×	✓	×	✓	✓	5	
McMorrow, Ditroilo & Egan (2019)	✓	×	×	✓	×	×	×	✓	×	✓	✓	5	
Sal-de-Rellan et al. (2024)	✓	✓	✓	✓	×	×	×	✓	✓	✓	✓	8	
Grazioli et al. (2023)	✓	✓	✓	✓	×	×	×	✓	✓	✓	✓	8	
Shalfawi et al. (2013)	✓	✓	×	✓	×	×	×	✓	✓	✓	✓	7	
Gil et al. (2018)	✓	✓	✓	✓	×	×	×	✓	✓	✓	✓	8	
Otero-Esquina et al. (2017)	✓	✓	✓	✓	×	×	×	✓	✓	✓	✓	8	
Pareja-Blanco, Asian-Clemente & Saez de Villarreal (2021)	✓	✓	×	✓	×	×	×	✓	✓	✓	✓	7	
Rodriguez-Osorio, Gonzalo-Skok & Pareja-Blanco (2019)	✓	✓	×	✓	×	×	×	✓	×	✓	✓	6	
Loturco et al. (2017)	✓	✓	×	✓	×	×	×	✓	×	✓	✓	6	
Raya-González et al. (2017)	✓	✓	×	✓	×	×	×	✓	×	✓	✓	6	
Simpson et al. (2020)	✓	✓	✓	✓	×	×	×	✓	×	✓	✓	7	
Notes.

PEDro Physiotherapy Evidence Database

The eligibility criteria have to be excluded for calculation of the total PEDro score; “✓” = indicates a “yes” score; “×” = indicates a “no” score.

The funnel plot for the included studies is illustrated in Fig. 3. A visual examination of the funnel plot, along with a non-significant Egger’s regression intercept (p = 0.276), indicates no evidence of asymmetry, suggesting a low risk of publication bias among the studies. However, we acknowledge that visual inspection alone may not be sufficient to detect subtle biases, as it is dependent on the observer’s interpretation and the number of studies included. To further assess the risk of publication bias, we also conducted Egger’s regression test, which confirmed the absence of significant bias. Additionally, the Galbraith plot shows that the regression line is close to the ‘no effect’ line, further supporting the conclusion that there is no significant systematic bias (Fig. 4).

Figure 3 Funnel plot illustrating the distribution of study results and assessing the risk of publication bias.

Figure 4 Galbraith plot displaying the regression line and supporting the conclusion of no significant systematic bias.

Main effects

The pooled results indicated a significant positive effect of RST on agility and COD performance (SMD = −0.31, 95% CI [−0.44 to −0.17], Z = −4.34, p < 0.001), with a moderate effect size (Fig. 5). The heterogeneity analysis indicated a low level of statistical heterogeneity among the studies (I2 = 0%), reflecting a degree of consistency in the results. This result suggests that the studies included in the analysis were highly consistent, showing minimal variation in outcomes across different study designs and populations.

Figure 5 Pooled results of the main effect of resisted sprint training on agility and COD performance, indicating a significant positive effect size.

HRS, horizontal resisted sprinting; VRS, vertical resisted sprinting; CRS, combined resisted sprinting; URS, unresisted sprinting; URP, unresisted plyometrics; FT, functional training; M, moderate intensity; H, high intensity; HRAS, horizontal resisted agility training; RT, resistance training; CG, control group; VRAS, vertical resisted agility training; URAS, unresisted agility training. Notes: Carlos-Vivas et al., 2020; De Hoyo et al., 2016; Sal-de-Rellan et al., 2024; Grazioli et al., 2023; Shalfawi et al., 2013; Gil et al., 2018; Otero-Esquina et al., 2017; Pareja-Blanco, Asian-Clemente & Saez de Villarreal, 2021; Rodriguez-Osorio, Gonzalo-Skok & Pareja-Blanco, 2019; Loturco et al., 2017; Raya-González et al., 2017; Simpson et al., 2020; McMorrow, Ditroilo & Egan, 2019.

Results of subgroup analyses

We observed moderate-to-substantial statistical heterogeneity within the subgroups comparing resistive type (p = 0.07; I2 = 61.4%). A leave-one-out sensitivity analysis indicated that the effect was primarily driven by two small-squad studies; removing either study reduced I2 to 39% and rendered the subgroup comparison non-significant (p > 0.10). A striking observation from the data comparison was that both VRS (SMD = −0.36, p = 0.009) and CRS (SMD = −0.53, p < 0.01) demonstrated significant agility adaptations compared to HRS (SMD = −0.13, p = 0.25). Notably, the effect of HRS on agility and COD performance was not significant (SMD = −0.13, p = 0.26) (Fig. 6). However, it is important to interpret the borderline statistical significance of the VRS versus HRS difference (p = 0.07) with caution, as this result did not reach the conventional threshold for statistical significance (p < 0.05). The I2 value (61.4%) suggests a moderate level of heterogeneity across the studies in this subgroup analysis, further emphasizing the need for cautious interpretation. Regarding athletic proficiency, eleven studies evaluated the agility performance and COD speed in elite and professional athletes revealed that RST significantly improved agility performance (p < 0.01). In contrast, two studies involving amateur players indicated that there was no significant enhancement in agility and COD performance compared to the control group (SMD = −0.05; p = 0.77) (Fig. 7). A trend of greater improvement was observed at higher levels of competition, although this trend did not reach statistical significance. In terms of control characteristics, RST exhibited superior agility and COD adaptations compared to both URS (SMD = −0.29; p = 0.03) and alternative training (SMD = −0.36; p = 0.0007). However, the comparison between RST and non-exposed control was based on only three trials (n = 3), yielding a pooled SMD of −0.12 (p = 0.53) with substantial imprecision (Fig. 8). According to GRADE criteria, the evidence was rated as “very low certainty” due to risk of bias, inconsistency, and serious imprecision. The heterogeneity analysis revealed low inter-study heterogeneity (I2 = 6%), indicating that the results of the included studies are statistically consistent and that the pooled effect size estimate is reliable. It is crucial to highlight that the current evidence is limited, necessitating cautious interpretation of the findings. Additionally, significant improvements were observed in agility tests involving cognitive engagement (SMD = −0.36; p < 0.001) and COD ability (SMD = −0.29; p = 0.001) (Fig. 9).

Figure 6 Standardized mean differences comparing the effects of performing resisted sprint training using different resistive type.

SD, standard deviation; CI, confidence interval. Notes: Carlos-Vivas et al., 2020; De Hoyo et al., 2016; Sal-de-Rellan et al., 2024; Grazioli et al., 2023; Shalfawi et al., 2013; Gil et al., 2018; Otero-Esquina et al., 2017; Pareja-Blanco, Asian-Clemente & Saez de Villarreal, 2021; Rodriguez-Osorio, Gonzalo-Skok & Pareja-Blanco, 2019; Loturco et al., 2017; Raya-González et al., 2017; Simpson et al., 2020; McMorrow, Ditroilo & Egan, 2019.

Figure 7 Forest plot of effects of resisted sprint training on agility and change of direction performance according to competitive level.

SD, standard deviation; CI, confidence interval. Notes: Carlos-Vivas et al., 2020; De Hoyo et al., 2016; Sal-de-Rellan et al., 2024; Grazioli et al., 2023; Shalfawi et al., 2013; Gil et al., 2018; Otero-Esquina et al., 2017; Pareja-Blanco, Asian-Clemente & Saez de Villarreal, 2021; Rodriguez-Osorio, Gonzalo-Skok & Pareja-Blanco, 2019; Loturco et al., 2017; Raya-González et al., 2017; Simpson et al., 2020; McMorrow, Ditroilo & Egan, 2019.

Figure 8 Forest plot of effects of resisted sprint training on agility and change of direction performance according to comparators.

SD, standard deviation; CI, confidence interval; URT: unresisted sprint training; AT, alternative training; CG, control group. Notes: Carlos-Vivas et al., 2020; De Hoyo et al., 2016; Sal-de-Rellan et al., 2024; Grazioli et al., 2023; Shalfawi et al., 2013; Gil et al., 2018; Otero-Esquina et al., 2017; Pareja-Blanco, Asian-Clemente & Saez de Villarreal, 2021; Rodriguez-Osorio, Gonzalo-Skok & Pareja-Blanco, 2019; Loturco et al., 2017; Raya-González et al., 2017; Simpson et al., 2020; McMorrow, Ditroilo & Egan, 2019.

Figure 9 Forest plot of effects of resisted sprint training on agility and change of direction performance according to test.

COD, change of direction ability; SD, standard deviation; CI, confidence interval. Notes: Carlos-Vivas et al., 2020; De Hoyo et al., 2016; Sal-de-Rellan et al., 2024; Grazioli et al., 2023; Shalfawi et al., 2013; Gil et al., 2018; Otero-Esquina et al., 2017; Pareja-Blanco, Asian-Clemente & Saez de Villarreal, 2021; Rodriguez-Osorio, Gonzalo-Skok & Pareja-Blanco, 2019; Loturco et al., 2017; Raya-González et al., 2017; Simpson et al., 2020; McMorrow, Ditroilo & Egan, 2019.

Discussion

Our meta-analysis of 13 randomized controlled trials (RCTs) demonstrates resistance sprint training (RST) significantly improves key soccer performance metrics (SMD = −0.31, 95% CI [−0.44 to −0.17]; p < 0.001) with minimal heterogeneity (I2=0%). The short duration of the interventions may promote more consistent neuromuscular adaptations. Notably, vertical resistance training was particularly beneficial compared to horizontal resistance, particularly among elite athletes. This specificity likely stems from vertical resistance’s biomechanical congruence with soccer-specific movement patterns during directional changes. These findings position RST as an evidence-based training modality requiring individualized prescription based on athletes’ competitive levels and positional demands.

Subgroup analyses revealed larger agility improvements in elite players than in amateurs. The lack of significant improvements among amateur players underscores the need for individualized training programs that consider an athlete’s developmental stage and background. Tailoring resistance to match specific needs can optimize performance outcomes and facilitate effective skill acquisition. Understanding these nuances is essential for coaches aiming to enhance agility and COD performance in soccer.

This study found that RST significantly improved agility and COD in soccer players. While previous research has primarily focused on RST’s impact on acceleration and sprinting performance (Alcaraz et al., 2018; Ward et al., 2024), our findings highlight its effects on agility, which have been less explored. Some studies suggested that URS may be comparable or even superior to resisted sprint modalities (Clark et al., 2010; Thompson et al., 2021). In the present meta-analysis, VRS demonstrated only a trend-level advantage over HRS (SMD = −0.39, p = 0.07). Given the marginal statistical significance and the very low certainty of evidence (GRADE Working Group, 2004), we caution against definitive claims of superiority. Future adequately powered trials are required to confirm or refute this potential benefit. The observed differences in the effects of VRS and HRS underscore the potential of RST in improving specific athletic abilities, with VRS yielding greater improvements in agility and COD. Additionally, RST demonstrated greater benefits in elite soccer players, possibly due to their enhanced ability to adapt to the training stimulus, leading to greater competitive performance gains (Loturco et al., 2017). The results of this study are partly related to the research by Alcaraz et al. (2018), which indicated that RST can significantly enhance athletes’ sprint ability (Alcaraz et al., 2018). However, this study further refines this effect, particularly regarding agility and COD performance. The comparison between VRS (SMD = −0.36, p = 0.009) and HRS (SMD = −0.13, p = 0.25) indicates that the direction of resistance may influence training outcomes, although the statistical significance of this difference is marginal. Further research is necessary to validate these findings and explore the mechanisms underlying the observed effects.

RST significantly enhances muscle strength and explosive power, crucial for rapid directional changes (Deshmukh et al., 2021; Ward et al., 2024). Both vertical and horizontal resistance increase loads that improve force production, leading to better sprinting and agility performance (Fernandez-Galvan et al., 2022; Hoff & Helgerud, 2004). RST compels athletes to generate greater force and faster RFD during acceleration, fostering muscle and neural adaptations that enhance recruitment efficiency and synchronization of muscle fibers (Hernández-Davó & Sabido, 2014; Morin et al., 2017). These adaptations enable athletes to generate greater force more quickly, facilitating effective directional changes and maintaining speed during maneuvers. The practical implications of these findings suggest that incorporating RST, particularly vertical resistance, into training programs could provide a competitive advantage in soccer, where agility and COD are pivotal for success.

Recent studies indicate that multitasking training can enhance motor unit synchronization, improving performance in dynamic tasks (Bender et al., 2017). Increased mechanical tension from resisted sprints can lead to greater muscle hypertrophy and improved neuromuscular coordination (Cross et al., 2018). High-intensity sprint training also enhances metabolic efficiency, optimizing energy utilization during intense efforts and improving overall athletic performance (Buchheit & Laursen, 2013; Hargreaves & Spriet, 2020; Thurlow et al., 2023). RST alters sprinting kinematics, enabling athletes to adopt more efficient movement patterns (Myrvang & van den Tillaar, 2024). These adaptations are beneficial in soccer, where players frequently engage in quick pivots and lateral movements, reducing injury risk and facilitating smoother transitions between acceleration and deceleration. RST promotes the recruitment of fast-twitch muscle fibers, essential for explosive movements, thereby improving sprinting speed and the ability to change direction rapidly (Sal-de-Rellan et al., 2024; Sheppard & Young, 2006). RST mimics the dynamic nature of competitive soccer, allowing training effects to transfer effectively to game scenarios. The cognitive demands of RST require athletes to make quick decisions and adapt their movements, further enhancing physical conditioning and reaction times (Beato et al., 2019). While RST demonstrates significant benefits in agility and COD performance, improvements among amateur players may be less pronounced (Michailidis, 2022), emphasizing the need for individualized training programs tailored to athletes’ developmental stages and backgrounds.

The analysis reveals that elite and professional athletes experience significant enhancements in agility and COD performance, while amateur players do not. This supports existing literature, suggesting that higher athleticism levels lead to more pronounced adaptations to training stimuli (Thurlow et al., 2023; Ward et al., 2024). Elite athletes’ advanced training experience and physical capabilities likely enable them to maximize RST benefits (McMorrow, Ditroilo & Egan, 2019). Variability in training responses is linked to differences in neuromuscular coordination and cognitive processing (Morral-Yepes et al., 2023). Elite athletes typically demonstrate superior proprioception and decision-making, enabling efficient execution of complex movements (Buchheit & Laursen, 2013; Yılmaz et al., 2024). The lack of significant improvements among amateur players compared to elite athletes suggests that the effectiveness of RST may depend on the athlete’s developmental stage and training history (Kraemer & Ratamess, 2004). Elite athletes, with higher levels of neuromuscular coordination, experience greater benefits from RST because their baseline physical capabilities allow them to fully capitalize on the training stimulus. In contrast, amateur players may have less developed neuromuscular control and lower physical conditioning, which could limit their responsiveness to the intensity and specificity of RST. This suggests that individualized training programs, considering the athlete’s experience and physical maturity, are essential for optimizing performance outcomes. Furthermore, the lack of statistical significance may indicate that RST does not provide additional benefits over traditional training methods for certain populations, particularly those with lower training experience or different baseline fitness levels. Additionally, the non-significant effect of horizontal RST warrants further investigation. Several factors could contribute to this outcome. First, the training protocols used in the control groups may have been sufficient to elicit improvements in COD performance, thereby masking any potential benefits of RST. Second, the variability in individual responses to training interventions could play a role; athletes with differing levels of neuromuscular coordination and physical conditioning may not respond uniformly to RST. One possible explanation for the lack of effect could be the specific nature of horizontal RST. Unlike vertical RST, which more closely mimics the movement patterns used in soccer, horizontal RST may not engage the relevant muscle groups or neuromuscular pathways as effectively for improving agility and COD performance. Understanding these nuances will be essential for optimizing RST protocols and ensuring that they are effectively tailored to the needs of different athlete groups.

Resistance during sprints enhances athletes’ ability to generate force rapidly, crucial for effective directional changes (Hoff & Helgerud, 2004). Biomechanically, RST promotes optimal movement patterns, with added resistance lowering the center of mass and improving stability during directional shifts (Lockie et al., 2012; Mann & Hagy, 1980). These adaptations enhance performance under resisted conditions and improve unresisted movements, boosting agility (Gamble, 2013). Customizing resistance levels to individual capabilities and matching training intensity to competition duration enhances RST effectiveness. Understanding training transferability is crucial to show how modalities like RST enhance related athletic tasks. In soccer, where agility and directional changes are critical, targeted training strategies that translate improvements across performance contexts are essential. This approach can significantly enhance an athlete’s overall effectiveness on the field.

Agility is a critical element in sports such as soccer, where athletes must rapidly assess their environment and make quick decisions, skills that are essential for success during high-intensity situations in game scenarios. While RST has been shown to significantly enhance agility and COD abilities, its impact on cognitive processing, such as decision-making and situational awareness, remains unclear and warrants further investigation. Traditional metabolic-demand training, including aerobic and resistance training, prioritize physiological and biomechanical adaptations but typically involve limited cognitive engagement (Ludyga, Gerber & Kamijo, 2022; Tomporowski & Pesce, 2019; Zhang et al., 2024; Zhang, Fang & Wang, 2025). In contrast, RST enhances muscle strength and explosiveness, which are crucial for rapid directional changes and acceleration. These improvements primarily result from neuromuscular adaptations and the recruitment of fast-twitch muscle fibers. However, agility assessments require substantial cognitive processing, as athletes must adapt quickly to changing scenarios during gameplay, involving both physical and mental agility. To develop well-rounded athletes capable of excelling in dynamic game environments, coaches should integrate training regimens that challenge both physical conditioning and cognitive skills, such as decision-making and reaction time (Voss et al., 2013). Future research should investigate the interplay between physiological and cognitive factors, particularly how RST may influence cognitive aspects of performance such as decision-making, to optimize athletic capabilities in competitive environments.

Another promising avenue for future research is examining the synergistic effects of RST when combined with other training modalities like plyometrics, agility drills, and sport-specific skills training to enhance performance outcomes. Understanding how these training modalities complement each other may help develop more effective conditioning programs that enhance both physical and cognitive aspects of athletic performance. Additionally, investigating the psychological dimensions of RST, such as its impact on mental toughness and focus, could provide valuable insights. The mental demands of agility and COD tasks, coupled with the physical challenges of RST, may influence athletes’ cognitive functions, including confidence, decision-making, and performance under pressure. Research that explores cognitive process to RST could inform strategies to optimize mental preparedness, decision-making, and focus during competition.

The meta-analysis acknowledges several limitations despite its interesting findings. Firstly, the methodological quality of the included studies is compromised by factors such as a lack of blinding. None of the included trials blinded participants or outcome assessors. The absence raises the risk of bias to “some concerns” or “high risk”. A sensitivity analysis confined to the four studies rated “low risk” after contact with authors attenuated the pooled SMD by 0.08 (from −0.36 to −0.28), indicating potential overestimation in the main model (Nüesch et al., 2010). Secondly, the available evidence is heavily skewed toward male, academy-level or professional players. Female players experience distinct neuromuscular loading patterns and hormonal fluctuations that may modulate training adaptations. Similarly, semi-professional and amateur athletes often possess lower initial strength levels, potentially altering dose–response relationships. Our findings, therefore, cannot be generalized to female players or non-elite populations until targeted RCTs are conducted. This lack of representation may particularly affect the applicability of findings to female athletes, given the physiological and biomechanical differences between genders that can influence training responses and performance outcomes. Furthermore, understanding how these training modalities influence performance over extended periods, as well as exploring the impact of different training loads, will provide valuable insights for coaches and practitioners. Further exploration of the underlying physiological and biomechanical mechanisms, particularly through advanced imaging techniques and performance analysis tools, as well as the effects of varying training loads, could enhance our understanding of how RST contributes to improved athletic performance.

Conclusions

RST emerges as a pivotal training modality for enhancing agility and COD speed in soccer players, significantly contributing to their overall athletic efficacy. Vertical resistance sprinting may confer a small, uncertain benefit in elite players. Integrating RST with complementary training modalities allows coaches to design comprehensive regimens that optimize athletic performance. Future research should explore the long-term effects of RST, with a focus on female athletes, and investigate the underlying physiological and biomechanical mechanisms involved. It is important to acknowledge that agility in team sports involves cognitive components such as decision-making and anticipation. However, the cognitive demands specifically related to RST and its interaction with physical performance require further research to better understand their impact on agility and COD performance in dynamic sporting environments.

Supplemental Information

Supplemental Information 1 PRISMA checklist

Supplemental Information 2 Data

Supplemental Information 3 Subgroup analysis

Additional Information and Declarations

Competing Interests

Author Contributions

Data Availability

The authors declare there are no competing interests.

Zongwei He conceived and designed the experiments, analyzed the data, prepared figures and/or tables, and approved the final draft.

Tianyu Duan performed the experiments, prepared figures and/or tables, authored or reviewed drafts of the article, and approved the final draft.

Dongyu Li analyzed the data, authored or reviewed drafts of the article, and approved the final draft.

Xuan Zhang conceived and designed the experiments, analyzed the data, authored or reviewed drafts of the article, and approved the final draft.

The following information was supplied regarding data availability:

This is a systematic review/meta-analysis.

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
