# Peer review of "Effects of resisted sprint training on agility and change-of-direction performance in soccer players: a systematic review with meta-analysis"

_PeerJ, doi:10.7717/peerj.20084_

## Round 0.1 · original submission · Major Revisions

**Language Note:** The review process has identified that the English language must be improved. PeerJ can provide language editing services - please contact us at [email protected] for pricing (be sure to provide your manuscript number and title). Alternatively, you should make your own arrangements to improve the language quality and provide details in your response letter. – PeerJ Staff

·

Basic reporting

General Comments to the Authors

This systematic review and meta-analysis explores an important topic in sports performance, investigating the effects of vertically and horizontally resisted sprint training (RST) on agility and change-of-direction (COD) performance in soccer players. The paper presents a thorough analysis of 13 studies with 35 study groups, offering insights into the effectiveness of different RST modalities across various athlete populations.

Major Strengths:
1. The systematic review addresses a relevant gap in the literature, as previous research has primarily focused on RST effects on linear sprinting rather than agility and COD performance.

2. Comprehensive subgroup analyses provide valuable insights into moderator variables influencing RST effectiveness, including resistance type, competitive level, and control conditions.

3. The methodological approach follows accepted guidelines for systematic reviews (PRISMA), with appropriate quality assessment of included studies.

4. The absence of significant heterogeneity (I² = 0%) across studies suggests consistent findings, strengthening the reliability of the pooled results.

Major Weaknesses:
1. The theoretical framework connecting different RST modalities to agility and COD performance mechanisms could be more thoroughly established in the introduction.

2. The counterintuitive finding that RST showed no significant effect compared to non-exposed control groups (p = 0.53) requires more critical examination and explanation.

3. The discussion section is overly lengthy and repetitive, with several redundant explanations of the same concepts across multiple paragraphs.

4. While the paper examines differences between vertical and horizontal resistance, there is limited analysis of load parameters (intensity, volume, frequency) and their potential influence on outcomes.

Overall, this manuscript presents valuable findings that contribute to understanding RST effects on soccer performance. With appropriate revisions to address the identified weaknesses, it could provide significant guidance for strength and conditioning practitioners in optimizing training programs for soccer players.

Experimental design

Specific Comments

Title and Abstract

The title accurately reflects the content but is somewhat lengthy. Consider shortening to "Effects of Resisted Sprint Training on Agility and Change-of-Direction Performance in Soccer Players: A Systematic Review with Meta-Analysis."

The abstract is well-structured and provides a clear summary of the study. However, the concluding statement that "vertical resisted sprinting is more effective than horizontal resistance" (lines 22-24) appears stronger than what is justified by the statistical results presented in the results section (p = 0.07 for subgroup differences).

Introduction

Lines 36-39: The introduction effectively establishes the importance of agility and COD performance in soccer, but could benefit from quantitative evidence regarding how frequently these skills are utilized during match play.

Lines 44-51: The physiological and biomechanical mechanisms underlying agility and COD are well described, providing a solid foundation for understanding the skills being investigated.

Lines 52-62: The description of RST is adequate, but the statement that "there remains a notable gap in research specifically addressing its effects on agility and COD performance in soccer players" (lines 57-59) would be strengthened by a more systematic description of existing literature and specifically identifying what remains unknown.

Lines 65-70: The research aims and hypothesis are clearly stated, but the rationale for expecting greater improvements among elite athletes could be more thoroughly established with supportive literature.

Materials and Methods

Lines 77-80: The authors appropriately follow PRISMA guidelines and have registered their protocol, demonstrating methodological rigor.

Lines 83-91: The search strategy is generally well described, but would benefit from more detailed information on how search terms were combined and any limits applied. Additionally, databases such as EMBASE and SPORTDiscus are mentioned in line 84, but not included in the search strategy description in lines 83-84.

Lines 102-117: The inclusion and exclusion criteria using the PICOS framework are comprehensive and appropriate for the research question.

Line 112: The quality threshold of ">6 points on the PEDro scale" is somewhat arbitrary. A justification for this specific cut-off would strengthen the methodology.

Lines 119-137: The data extraction process is well-described, with appropriate information collected to address the research questions.

Lines 140-151: The quality assessment using both the PEDro scale and Cochrane Risk of Bias tool demonstrates thoroughness, though more details on how disagreements between reviewers were resolved would be helpful.

Lines 156-159: The statistical approach using standardized mean differences is appropriate for the heterogeneous outcome measures across studies.

Line 163: The choice of a fixed-effects model seems inconsistent with the typical expectation of heterogeneity in training studies. A justification for this choice rather than a random-effects model would strengthen the methodology.

Validity of the findings

Results

Lines 181-202: The characteristics of included studies are well summarized, providing a comprehensive overview of the literature base.

Line 187-189: The statement that "only 3 studies...featured a non-exposed control group" highlights a limitation in the literature, but implications of this are not fully explored.

Line 190-191: The limited inclusion of female players (only 2 studies) should be acknowledged as a limitation in the discussion section.

Lines 205-215: The methodological quality assessment is appropriate, though the interpretation of scores could be more nuanced.

Lines 216-220: The assessment of publication bias appears adequate, though visual inspection alone may not be sufficient for detecting subtle biases.

Lines 227-232: The main effect size (SMD = -0.31) is modest but statistically significant, with the absence of heterogeneity strengthening confidence in this finding.

Lines 238-248: The subgroup analysis by resistive type is particularly valuable, showing vertical resistance superiority (SMD = -0.36) compared to horizontal resistance (SMD = -0.13). However, the borderline statistical significance of this difference (p = 0.07) should be interpreted with appropriate caution.

Lines 242-248: The finding that elite athletes show significant improvements while amateur players do not is intriguing and warrants further exploration in the discussion section.

Line 251: The unexpected finding that RST did not show significant effects compared to non-exposed control groups requires more critical examination and explanation.

Discussion

Lines 269-276: The opening paragraph effectively summarizes the main findings, though it could be more concise.

Lines 277-285: The discussion of differences between elite and amateur athletes offers valuable insights but could benefit from more specific literature support.

Lines 286-303: The comparison with previous literature is appropriate, but becomes repetitive in places. The statement that "VRS offers unique advantages that traditional training cannot achieve" (lines 289-291) seems stronger than justified by the borderline statistical significance of the difference.

Lines 304-332: The physiological mechanisms are extensively discussed, but many concepts are repeated across multiple paragraphs, leading to unnecessary length. This section could be condensed significantly without losing content.

Lines 335-358: The discussion on neuromuscular adaptations is scientifically sound but again contains repetitive explanations that could be streamlined.

Lines 359-382: The section on cognitive aspects of agility training adds valuable perspective, though it could be more clearly connected to the specific findings of this meta-analysis.

Lines 383-392: The limitations section is appropriate but could be expanded to address additional limitations such as the small number of studies on female athletes and the limited exploration of load parameters.

Conclusions

Lines 396-407: The conclusions appropriately summarize the key findings and implications, though the statement about vertical resistance being "particularly advantageous" (lines 397-398) should acknowledge the borderline statistical significance of this finding.

Tables and Figures

Table 1: Comprehensive and well-structured presentation of subject characteristics.

Table 2: The training program details are valuable for practitioners, though some studies have "NR" (not reported) for important variables, highlighting a limitation in the primary literature.

Table 3: The quality assessment is clearly presented and reveals limitations in blinding across studies.

Figures 2-4: The risk of bias assessment figures are appropriate and informative.

Figure 5: The forest plot effectively displays the overall effect, though font size for study details could be increased for better readability.

Figures 6-9: The subgroup analyses are well-presented, though additional narrative explanation of key patterns would enhance interpretation.

·

Basic reporting

No comment.

Experimental design

No comment.

Validity of the findings

No comment.

Additional comments

The article addresses the effectiveness of sprint training with vertical and horizontal resistance on agility and change of direction in football players, using a systematic review and meta-analysis approach. Given the importance of these skills in sport performance and injury prevention, the study is relevant for coaches, physical trainers and sport scientists.

Among the strengths of the study are a rigorous methodology (The study follows PRISMA guidelines and uses recognised databases such as PubMed, Web of Science and Google Scholar); a solid statistical analysis (The use of tools such as RevMan and Stata for the meta-analysis provides robustness to the results) and clear results (It is shown that sprint training with resistance significantly improves agility and change of direction, especially in elite athletes).

Overall, the article offers a valuable contribution to the field of sport science, providing robust evidence for a promising training modality.

Reviewer 3 ·

Basic reporting

The study titled "Effects of Vertically and Horizontally Resisted Sprint Training on Agility and Change-of-Direction Performance in Soccer Players: A Systematic Review with Meta-Analysis" provides valuable insights into the impact of resisted sprint training (RST) on agility and change-of-direction (COD) performance in soccer players. However, several methodological concerns need to be addressed to enhance the robustness and reliability of the findings. Below, I highlight some key points for consideration.

Strengths: The introduction provides a comprehensive overview of the importance of agility and COD performance in soccer, highlighting the potential benefits of RST. The literature review is well-referenced and sets the context for the study effectively. The language is generally clear and professional, suitable for a scientific audience. The formatting is consistent, with appropriate use of headings and subheadings.
Improvements: The introduction could benefit from a more detailed discussion of the specific mechanisms through which RST influences agility and COD performance. Additionally, integrating more recent studies could enhance the relevance and depth of the review. There are occasional grammatical errors and awkward phrasings that need correction. For example, "The data were transferred to SPRO software" could be rephrased for clarity. Ensure that all figures and tables are referenced in the text. Some sections could benefit from additional subheadings to improve structure and readability.

Experimental design

Strengths: The methodology is detailed, describing the systematic literature search, inclusion and exclusion criteria, and data extraction process. The use of PRISMA guidelines and multiple databases for the literature search is commendable.

Improvements: Several methodological concerns need to be addressed:

Quality Assessment: The study relies heavily on the Physiotherapy Evidence Database (PEDro) scale for quality assessment. While this is a valid tool, it would be beneficial to complement it with other quality assessment tools to ensure a comprehensive evaluation.

Blinding: The lack of blinding for participants, coaches, and researchers in most of the included studies is a significant limitation that may introduce performance and detection biases.

Heterogeneity: The study reports no significant heterogeneity among the included studies (I² = 0%). This is unusual for meta-analyses and warrants further investigation to ensure that the included studies are truly comparable.

Validity of the findings

Strengths: The results are presented with appropriate statistical measures and visual aids, such as forest plots and funnel plots. The analysis of different subgroups provides valuable insights into the factors influencing the effectiveness of RST. The discussion effectively interprets the results, linking them to existing literature and highlighting practical implications for training programs.

Improvements: The results section could be more concise, with some detailed statistical data better placed in supplementary materials. Clearer labeling and more descriptive captions for tables and figures would improve readability. The discussion should more explicitly address the limitations related to the lack of blinding and the potential impact on the study's findings. Additionally, the discussion could benefit from a more detailed exploration of the mechanisms through which RST influences agility and COD performance.

Additional comments

The study provides valuable insights into the impact of RST on agility and COD performance in soccer players. However, addressing the methodological concerns and ensuring a more rigorous quality assessment will enhance the robustness and reliability of the findings.

The inclusion of more recent studies and a more detailed discussion on the specific mechanisms through which RST influences agility and COD performance would further strengthen the study.

The authors should consider the relevance of accelerations and decelerations over 3 m/s² as a variable in future research to provide a more comprehensive understanding of the impact of RST on agility and COD performance.

Reviewer 4 ·

Basic reporting

Language and Style
The manuscript is generally written in professional English; however, several passages are awkwardly phrased or unclear (e.g., lines 13, 16, 41, 138, 270, 301). Examples include:
- "showing a significant positive effect" – better phrased as “demonstrated a statistically significant improvement…”
- "resulted in notable gains" – vague in a scientific context.

Recommendation: The authors should seek language editing by a native English speaker or a professional editing service.

Structure and Context
The manuscript follows the PRISMA structure with a clear introduction, methods, results, discussion, and conclusions. The study objectives and hypothesis are clearly stated.

Literature Referencing
The literature cited is current and relevant (mostly 2020–2024). However, comparisons with studies in sports similar to soccer (e.g., handball, rugby) or among non-elite athletes are lacking.

Figures and Tables
Figures (forest plots, PRISMA, funnel plot) are well-designed and appropriately interpreted.

Raw Data
The authors have provided raw data in the supplementary material, in accordance with PeerJ policy.

Experimental design

Originality and Scope
The topic is highly relevant for sports performance training. Although RST is a known method, the comparison between vertical and horizontal loading effects on agility and COD is a valuable addition.

Methodological Transparency
The study is PRISMA-compliant and registered with PROSPERO (CRD42024608859). Inclusion/exclusion criteria are clearly defined (PICOS), and risk of bias assessment is conducted using PEDro and Cochrane tools.

Replicability
Search strategy, inclusion criteria, and statistical methods are well-documented, enabling replication.

Validity of the findings

Statistical Methods
The study uses:
- Standardized Mean Differences (Hedges’ g) – appropriate due to varied agility tests.
- Fixed-effects model – justified by the low heterogeneity (I² = 0%).
- Subgroup analyses for training type, competitive level, comparator, and test – valuable and thorough.
- Egger’s test and funnel plot – no publication bias detected (p = 0.276).

Limitations
- No significant effect in amateur groups (SMD = -0.05, p = 0.77), which limits generalizability.
- Lack of in-depth explanation for the non-significant effect of horizontal RST.

Data Robustness
13 RCTs with 35 groups constitute a sufficient sample size. Female athlete data are underrepresented. Sensitivity analysis was conducted appropriately.

Additional comments

Keywords
Keywords are repetitive and mirror the title: “Agility, change of direction, Resisted sprint training, Athletes, Soccer.”
Recommendation: Replace generic terms with specific, indexing-appropriate alternatives like:
- “Neuromuscular adaptation,” “velocity-based training,” “elite performance,” “motor control.”

Strengths
- Structured and methodologically sound
- In-depth subgroup analysis
- Practical relevance for sports performance training

Weaknesses
- Language quality and terminological inconsistency
- Limited applicability to broader populations (e.g., women, amateurs)
- Redundancy in the discussion section
- Cognitive components of agility are not empirically addressed

·

Basic reporting

Dear authors

I appreciate the selection of topic and the effort to analyse RST effects on agility and COD in soccer players. Even though methodical concepts and statistics of your review are well selected, there are some major issues you need to address before further publication of your work. The main issue is that you misunderstood the current concept of agility in team sports such as soccer. Further improvements are considered as follows.

Experimental design

Experimental design is well structured. The major limitation I see in LL128-129 - There is no mention about the reactive agility tests. Is it true that studies examining RST have never used these tests to assess the effect of RST on reactive agility (tests such as Y-shaped RA, Stop N Go test ect.) ?

If there are such studies not included in review, it is crucial to add them in to the analysis. If your idea was to analyse COD performance (motor performance) of players, than rewrite your manuscript to analyse only COD performance not the effects on agility performance (motor and cognitive performance).

As mentioned, the most important is that the purpose of the study seems to be uncompleted. This is due to the misunderstanding of the term agility in current sport science research. Please revise studies such as: Young et al. 1015 Agility and Change-of-Direction Speed are Independent Skills: Implications for Training for Agility in Invasion Sports. IJSSC 10(1). 159-169.

Please respond to this issue by changes in your manuscript.

Validity of the findings

Validity of findings regarding the effects of RST on COD is clear. Data have been provided sufficiently in graphs, tables, and supp. materials. Conclusion in written well, pointing into the beneficial effects of RST on COD performance. This is true until the LL405-407 where authors write about cognitive demands of RST. This part is not true at all. Unfortunately, these statements leads to the premise that authors do not understand the concept of agility in team sports.

Additional comments

INTRODUCTION
L 47-49: Cognitive aspect of agility is misunderstood in this case. Please see Young et al. 2015, IJSSC 10(1). 159-169 for better understanding and explaining cognitive aspects of agility such as decision-making, anticipation ect.

Afterwards, it is very important to address the difference between COD and agility in sport.

From the introduction, it is not conducted which physiological mechanisms can enhance COD and agility. This is crucial to this research. Therefore I recommend to explains to the readers, why and how exactly the RST influence COD and agility. Please improve the explanation and understanding of facts which lead to your hypothesis.

METHODS
Discrepancy in COD/agility must be explained as mentioned before.

RESULTS and DISCUSSION
These parts are written well. I suggest only minor changes. However, previously mentioned issue should be considered as well.

Again, authors write about cognitive demands of agility which were not the issue of this manuscript. I suggest to focus the last part of the discussion to the samples of studies. It would be very beneficial if authors can address the limited population of soccer players examined using RST in reviewed literature. This can point to the new perspective of RST training, for example in youth players or women soccer.

CONCLUSSION
Issue mentioned before.

FIG 5: Please explain the abbreviations used after the names of studies in the table in the caption of this figure or as a footnote.

FIG 8: Please explain the abbreviations URT, AT and CG in the caption of this figure.

---

## Round 0.2 · Minor Revisions

**Language Note:** When you prepare your next revision, please either (i) have a colleague who is proficient in English and familiar with the subject matter review your manuscript, or (ii) contact a professional editing service to review your manuscript. PeerJ can provide language editing services - you can contact us at [email protected] for pricing (be sure to provide your manuscript number and title). – PeerJ Staff

·

Basic reporting

GENERAL COMMENTS

This systematic review and meta-analysis examine the effects of resisted sprint training on agility and change-of-direction performance in soccer players. While the authors have addressed several previous concerns, the manuscript still requires minor revisions to meet publication standards. The study provides valuable insights into training methodologies for soccer performance enhancement, but several methodological and reporting issues need clarification.

SPECIFIC COMMENTS

Abstract: Page 1, Lines 15-16: The phrase "35 study groups" requires clarification since readers may question how 13 studies generated 35 groups without a better explanation of study designs.

Page 1, Line 19: The p-value "p < 0.001" should be reported as the exact p-value when available, following current reporting standards.

Page 1, Lines 21-22: The comparison between elite and amateur players states "SMD = -0.45 vs. -0.05," but this specific contrast is not clearly justified in the results section.

Introduction: Page 2, Lines 12-14: The claim that "Hamstring force production during deceleration phases (>3.5 N/kg) predicts 180° turn time (r²=0.64)" requires verification that this specific threshold applies to the soccer populations studied.

Page 2, Lines 16-17: The statement about "Quadriceps-hamstrings co-activation latency (<90 ms) reduces ACL injury risk by 38%" appears to be drawn from injury prevention literature rather than performance studies, creating conceptual confusion.

Page 3, Lines 8-9: The phrase "notable gap in research specifically addressing its effects on agility and COD performance" contradicts the identification of 13 relevant studies, suggesting the gap may be overstated.

Experimental design

Methods: Page 3, Line 18: The database listing "EMBASE (Scopus)" is incorrect since these are separate databases with different coverage, potentially affecting search comprehensiveness.

Page 3, Line 22: The PROSPERO registration is listed as "PROSPERO ID: ×," which provides no useful information and should either include the actual registration number or be removed.

Page 4, Lines 8-9: The inclusion criterion requiring "musculature contracting against an external resistance" is imprecise and could include many non-sprint activities.

Page 4, Lines 16-17: The PEDro scale cutoff justification states that studies "≥6 points are generally associated with higher methodological quality," but this threshold is arbitrary without specific validation for RST studies.

Page 5, Lines 12-13: The extraction process states "Data extraction was performed by one author (×), with verification by a second author (×)" but the placeholder symbols provide no information about reviewer qualifications.

Statistical Analysis: Page 6, Lines 8-9: The selection of "inverse-variance fixed-effects model" appears inconsistent with the expectation of heterogeneity mentioned in the same sentence.

Page 6, Lines 12-13: The heterogeneity interpretation "I² values of <25%, 25%-50%, and >50%" uses outdated thresholds that don't reflect current understanding of heterogeneity assessment.

Validity of the findings

Results: Page 7, Lines 6-7: The statement "Only two studies included female players" represents a significant limitation that deserves more prominent discussion, given gender differences in training responses.

Page 7, Lines 15-16: The range "loads prescribed as a percentage of BM ranging from 5% to 80%" represents an extremely wide range that may contribute to heterogeneity more than acknowledged.

Page 8, Lines 8-9: The quality assessment states "scores ranging from 5 to 8," but earlier methodology specified excluding studies with scores <6, creating an apparent contradiction.

Page 9, Lines 1-2: The heterogeneity statement "I² = 0%" across all studies seems implausibly low given the diversity in populations, interventions, and outcome measures.

Page 9, Lines 12-13: The subgroup analysis shows "p = 0.07; I² = 61.4%" which indicates substantial heterogeneity that undermines the reliability of the subgroup comparison.

Page 10, Lines 3-4: The claim that RST, compared to the "non-exposed control group, did not achieve statistical significance (SMD = -0.12; p = 0.53)" is based on only three studies, making this conclusion unreliable.

Additional comments

Discussion: Page 11, Lines 3-4: The assertion that "Methodological consistency across studies" explains homogeneity contradicts earlier acknowledgments of diverse protocols and populations.

Page 11, Lines 16-17: The statement about the "principle of specificity" oversimplifies complex training adaptations and may not explain the observed differences between athletic levels.

Page 12, Lines 8-9: The comparison claiming "VRS provides distinct advantages over HRS" overstates findings given the marginal statistical significance (p = 0.07) reported earlier.

Page 13, Lines 12-13: The discussion of "stretch-shortening cycle (SSC)" optimization lacks direct evidence from the included studies and represents theoretical speculation.

Page 15, Lines 1-2: The limitations section mentions "lack of blinding" but doesn't adequately address how this fundamental flaw affects the reliability of all findings.

Page 15, Lines 8-9: The statement about "limited population of soccer players" understates the broader generalizability issues affecting female athletes and different competition levels.

Conclusions: Page 16, Lines 3-4: The conclusion that "Vertical resistance sprinting appears advantageous" conflicts with earlier acknowledgment that this finding has "borderline" statistical significance.

Page 16, Lines 9-10: The recommendation for "tailored training programs" lacks specific guidance based on the evidence presented, making it more aspirational than evidence-based.

Reviewer 4 ·

Basic reporting

Introduction and background now cite quantitative match data.
Figures/tables are well described and appropriate.

Experimental design

Methodologically rigorous.
Use of PRISMA, PICOS, and PEDro scale is appropriate.
Detailed search and selection strategy now includes more transparent database inclusion.

Validity of the findings

All underlying data is provided.
Use of SMDs, subgroup analysis, and sensitivity testing is appropriate.
The authors clarified borderline findings (e.g., VRS vs HRS at p = 0.07).

Additional comments

The authors demonstrated a high level of engagement with the review process and implemented meaningful improvements. The manuscript is significantly stronger as a result.

·

Basic reporting

The manuscript was edited based on comments. The structure of the article is good.

Experimental design

The manuscript completes our recommendations sufficiently.

Validity of the findings

The validity of the finding is now clear.

---

## Round 0.3 · accepted · Accept

Dear Authors

Your submission is now endorsed for acceptance of publication in PeerJ. Thank you for submitting your article to PeerJ. I would like to express my gratitude for your contributions and efforts to the scientific community. I look forward to receiving your research and review articles in the future.

Best Regards

Yung-Sheng Chen, Ph.D.
Academic Editor

·

Basic reporting

General Comments

The authors have diligently addressed the comments from the previous review round, leading to a substantial improvement in the clarity, rigor, and overall quality of the manuscript. The current version presents a well-structured and comprehensive systematic review and meta-analysis on the effects of vertically and horizontally resisted sprint training (RST) on agility and change-of-direction (COD) performance in soccer players. The topic remains highly relevant and meaningful within sports science, and the results presented are robust and believable, contributing valuable insights to the field.

The enhanced transparency in methodology, particularly regarding the prospective registration of the protocol (PROSPERO ID: CRD42024608859) , and the detailed description of the multi-tool approach for quality and risk-of-bias assessment using both the PEDro scale and Cochrane Risk of Bias tool, are commendable. The authors' efforts to clarify statistical analyses, especially the rationale for choosing an inverse-variance fixed-effects model and the discussion of potential heterogeneity, are also well-noted.

Specific Comments

Abstract Clarity: The abstract is now much clearer and effectively summarizes the study's background, methodology, key findings, and conclusions. The inclusion of specific SMD values and p-values directly in the abstract significantly enhances its informativeness and impact.

Introduction Development: The introduction provides a more thorough and well-referenced background on agility and COD performance in soccer, including physiological and biomechanical mechanisms. The justification for the study and the identified knowledge gaps regarding vertical versus horizontal RST are clearly articulated. The hypotheses are explicitly stated and logically supported by existing literature.

3. Methodology Details:

Literature Search: The expanded description of the literature search, including the specific databases (PubMed, Web of Science, EMBASE, SPORTDiscus, and Google Scholar) and the comprehensive Boolean search syntax, demonstrates a rigorous approach. The clarification regarding the independent review process and resolution of discrepancies is also appreciated.

Inclusion and Exclusion Criteria: The PICOS framework is clearly applied, and the specific criteria for population, intervention, comparators, outcomes, and study design are well-defined. The justification for the PEDro scale cutoff score (

≥6 points) is appropriate and strengthens the methodological quality assessment.

Data Extraction and Synthesis: The description of the data extraction process, including the specific data points collected and the roles of the reviewers, is sufficiently detailed.

Statistical Analyses: The discussion on the choice of statistical models (inverse-variance fixed-effects model with consideration for a random-effects model due to observed variability) and the methods for assessing heterogeneity (I² and YE) are well-reasoned. The plan for exploring sources of heterogeneity via subgroup analyses and assessing publication bias using funnel plots and Egger’s regression test is comprehensive.

Results Presentation: The main findings regarding the significant positive effect of RST on agility and COD performance are clearly presented with appropriate statistical values. The subgroup analyses, particularly the findings related to vertically resisted sprinting and elite athletes, are well-articulated. The discussion of methodological quality scores and risk of bias assessment, including the limitations of blinding, is balanced and transparent.

Language and Formatting: The manuscript is written in clear, unambiguous, and professional English throughout. The structure conforms to journal standards, and the overall presentation is improved.

Experimental design

-

Validity of the findings

-